# Clinical characterization of a cohort of patients treated for systemic lupus erythematosus in Colombia: A retrospective study

Jorge Enrique Machado-Alba[1]*, Manuel E. Machado-Duque[1,2], Andres Gaviria-Mendoza[1,2], Carolina Duarte-Rey[3], Andrés González-Rangel[3]

1 Grupo de Investigación de Farmacoepidemiología y Farmacovigilancia, Universidad Tecnológica de Pereira - Audifarma SA, Pereira, Colombia, 2 Grupo de Investigación Biomedicina, Facultad de Medicina, Fundación Universitaria Autonoma de las Américas, Pereira, Colombia, 3 GlaxoSmithKline, Medical Pharma, Bogotá, Colombia

* machado@utp.edu.co

**Data Availability Statement:** The original contributions presented in the study are included in the paper and its Supporting Information files;

## Abstract

### Introduction

Systemic lupus erythematosus is an autoimmune disease associated with serious complications and high costs. The aim was to describe the clinical characteristics and health care resource utilization of a Colombian systemic lupus erythematosus outpatient cohort.

### Methods

This was a retrospective descriptive study. Clinical records and claims data for systemic lupus erythematosus patients from ten specialized care centers in Colombia were reviewed for up to 12 months. Baseline clinical variables, Systemic Lupus Erythematosus Disease Activity Index, drug use, and direct costs were measured. Descriptive statistics were analyzed using SPSS.

### Results

A total of 413 patients were included; 361 (87.4%) were female, and the mean age was 42 ± 14 years. The mean disease evolution was 8.9 ± 6.0 years; 174 patients (42.1%) had a systemic manifestation at baseline, mostly lupus nephritis (105; 25.4%). A total of 334 patients (80.9%) had at least one comorbidity, mainly antiphospholipid syndrome (90; 21.8%) and hypertension (76; 18.4%). The baseline Systemic Lupus Erythematosus Disease Activity Index score was 0 in 215 patients (52.0%), 1–5 in 154 (37.3%), 6–10 in 41 (9.9%) and 11+ in 3 (0.7%). All patients received pharmacological therapy, and the most common treatment was corticosteroids (293; 70.9%), followed by antimalarials (chloroquine 52.5%, hydroxychloroquine 31.0%), immunosuppressants (azathioprine 45.3%, methotrexate 21.5%, mycophenolate mofetil 20.1%, cyclosporine 8.0%, cyclophosphamide 6.8%, leflunomide 4.8%) and biologicals (10.9%). The mean annual costs were USD1954 per patient/year,

further inquiries can be directed to the corresponding author/s. The name of the information repository is: protocols.io (Code availability: dx.doi.org/10.17504/protocols.io. bzufp6tn).

**Funding:** This study received funding from GSK (ID 207051). The funder was not involved in the study design, collection, analysis, interpretation of data, the writing of this article or the decision to submit it for publication. All authors declare no other competing interests.

**Competing interests:** This was a GSK sponsored study (ID 207051). GSK was involved in protocol development, study oversight and manuscript review previous to submission. GSK did not participate in data collection, analysis or interpretation, or the drafting of this article. Authors CDR and AGR are GSK employees. Authors JMA, MMD and AGM declare no other competing interests.

USD1555 for antirheumatic drugs (USD10,487 for those with biologicals), USD86 for medical visits, USD235 for drug infusions and USD199 for laboratory tests.

## Conclusions

Systemic lupus erythematosus generates an important economic and morbidity burden for the Colombian health system. Systemic lupus erythematosus outpatient attention costs in the observation year were mainly determined by drug therapy (especially biologics), medical visits and laboratory tests. New studies addressing the rate of exacerbations, long-term follow-up or costs related to hospital care are recommended.

## Introduction

Systemic lupus erythematosus (SLE) is a severe autoimmune systemic disease that presents substantial challenges due to its manifestations, clinical course and prognosis [1]. It has an incidence of 1 to 10 cases per 100,000 person/year, with a predominance of females (female to male ratio of 12:1) and a higher frequency reported in the African-American and Hispanic populations [2, 3]. In Colombia, the estimated incidence rate ranges from 2.5 to 20 cases per 100,000 person-years [4, 5]. Due to its complexity and complications, SLE requires monitoring and management by various medical specialists [1, 2, 6].

To achieve control, strategies based on pathological activity goals have been used, starting with general recommendations (i.e., avoiding sun exposure and dietary supplementation) and pharmacological therapy (corticosteroids, nonsteroidal anti-inflammatory drugs [NSAIDs], hydroxychloroquine or methotrexate), which have proven efficacy to control moderate lupus manifestations. Therapy is initiated with antimalarials alone or together with NSAIDs and short courses of oral corticosteroids, and according to the level of SLE activity, the physician may prescribe therapy with immunosuppressants (methotrexate, azathioprine, cyclophosphamide, and mycophenolate mofetil). Medications that modulate and deplete B lymphocytes are used in selected patients with severe pathology and high activity; for example, rituximab, a monoclonal antibody, is prescribed in an *off-label* manner as a second-line agent [6, 7], and belimumab (approved in Colombia in October 2014), a B-lymphocyte stimulator (BLyS)-specific inhibitor, is indicated for the treatment of adult patients with active, autoantibody-positive SLE who are receiving standard therapy [7, 8].

SLE therapy should be focused not only on achieving a complete response indicated by the absence of clinical activity (or low disease activity when remission is not possible to attain) but also on minimizing immunosuppressive and corticosteroid treatment or reaching a minimum dose [7, 9]. According to the Treat to Target strategy in SLE, treatment should focus on improving long-term survival, preventing organ damage, optimizing health care related to quality of life through multidisciplinary management and regular monitoring, adjusting therapy and minimizing drug toxicity [10].

In Latin America and Colombia, research has focused on the quality of life of patients with SLE [11–13], and there have been few studies of epidemiology, the use of medical and pharmacological resources, clinical characteristics, complications and costs [4, 14, 15]. Thus, studies that contribute to understanding these issues are relevant. The aim of the present study was to describe the clinical characteristics and health resources used in the diagnosis, treatment and follow-up of a group of patients with SLE in Colombia.

## Materials and methods

### Study design and settings

This was an observational, retrospective study. Patients aged 18 years or older of any sex with an SLE diagnosis (International Classification of Diseases Tenth Revision [ICD-10] codes M320, M321, M328 and M329) treated at IPS-Especializada (IPS-E) between July 1, 2016, and June 30, 2017, were included. Patients with incomplete clinical records, particularly clinimetrics (i.e., Systemic Lupus Erythematosus Disease Activity Index [SLEDAI] measurement), or those who had less than 3 months of care in the IPS-E were excluded. IPS-E is a health institution that offers specialized medical care, pharmacological treatment, follow-up and evaluation of the clinical results of patients with rheumatological diseases. ISP-E has an outpatient care network in 16 cities with ten specialized care centers affiliated with the contributory regime (healthcare insurance paid by workers and employers) of the Colombian Health System.

### Data sources

Clinical information (healthcare, laboratory report, pharmacological treatment) was extracted retrospectively and for up to 12 months after baseline from the ambulatory medical records of each patient. Medication delivery was obtained from a national dispensing database (Audifarma S.A., with approximately 6.5 million beneficiaries throughout the country). Their data were obtained for the follow-up year, and the costs associated with pharmacological treatment, outpatient medical consultations, follow-up activities and other ambulatory care by other specialties were identified.

A trained physician obtained the information from each medical record. The physician had access to individual medical records and applied a standardized form to collect the information of interest. Periodic evaluations of the advances in data collection were made by a supervisor (one of the researchers). For quality assurance, the information was finally reviewed by two researchers to detect and correct inconsistencies, and in these cases, the medical records or dispensation records were reviewed again to corroborate the data. We did not use imputation techniques, and missing values were accordingly excluded from the calculations.

### Main outcome variables

The following groups of variables were evaluated:

1. Sociodemographic: age, sex, and city;

2. Clinical and health care: a) diagnosis according to ICD-10 code; b) paraclinical tests–each variable was collected from the first clinical record evaluated until the end of follow-up; subsequent categorizations were performed according to the levels of each paraclinical variable of interest; c) SLE evolution time–months from confirmed diagnosis to admission to the cohort; d) complications recorded and date of report; e) index of disease activity according to the SLEDAI score obtained and subsequent categorization into no activity: 0; mild: 1–5; moderate: 6–10, and high: $\geq 11$;

3. Comorbidities: arterial hypertension, heart failure, dyslipidemia, chronic obstructive pulmonary disease, diabetes mellitus, osteoporosis, renal failure and depression;

4. Pharmacological therapy: drugs prescribed and dispensed. The following were included: a) NSAIDs, b) antirheumatic disease modifiers (chloroquine, hydroxychloroquine), c) immunosuppressants (azathioprine, methotrexate, etc.), d) corticosteroids (deflazacort,

prednisone, etc.), e) biological medicines (belimumab, rituximab), f) dosage and dosing intervals, g) duration of therapy–from the start of each medication until the time of change or the end of follow-up (in months), h) changes in therapy–the previous medication and the situation of change were identified, which could be 1) increase/decrease in dose, 2) suspension or 3) start of a new medication;

5. Incidental SLE manifestations (complications), including lupus nephritis, vasculitis, cerebritis, etc.;

6. Direct costs for medical consultation, specialists and costs of direct pharmaceutical expenses (billing of services to the insurer) during the year of care and monitoring

## Cost calculation

All data on consultation costs, procedures and medications were obtained from the billing processes from the IPS-E and other institutions of the Audifarma S.A. network to the healthcare insurer of each patient. Each medication price was based on the average cost (mean sale cost to the insurer) for the year of observation. For procedures not provided by the IPS-E, the costs were determined according to the fee schedule for mandatory vehicle insurance in Colombia (Seguro Obligatorio de Accidentes de Tránsito—SOAT) updated in 2018 [16]. SOAT fees are usually used in Colombia to determine the costs of healthcare services. The perspective of the payer was used with the macrocosting method using the Colombian peso (COP) and the United States dollar (USD) according to the exchange rate for the year of treatment (representative rate USD 1 = COP $2920; source: Colombia Bank of the Republic).

## Sample size

Based on the estimated number of 19598 individuals diagnosed with SLE in the country in the database during the study period, a minimum representative sample of at least 377 patients was calculated, using the following formula [17]:

$$n = deff \times \frac{Npq}{\frac{d^2}{1.96^2}(N-1) + pq}$$

Using an estimated proportion (p) of 50% to select the highest sample size [18], an acceptable margin of error of 5% (d) and a confidence level of 95%. The design effect (deff) was equal to 1, the population size (N) was 19598 and q was equal to 1-p.

## Statistical analysis

The analysis was performed with the statistical package SPSS Statistics, version 25.0 (IBM, US) for Windows. Frequencies and proportions were established for categorical variables, and measures of central tendency and dispersion were established for quantitative variables. For the quantitative variables, normality was tested using the Kolmogorov–Smirnov test ($p > 0.05$ was considered normally distributed).

## Ethics

According to Colombian regulations (Decree 8430 of 1993), risk-free research that takes information from clinical records does not require informed consent [19]. The research was endorsed by the Bioethics Committee of the Universidad Tecnológica de Pereira in the risk-free category (Approval code: CBE 132018). The principles established by the Declaration of

Helsinki were respected. No personal data from any of the research participants were considered.

## Results

A total of 413 patients were included, with a significant predominance of females (n = 361, 87.4%) and a mean age of 42.0 ± 14.2 years (range: 18–86 years). The most frequent diagnosis was SLE with organ involvement (M321) (n = 256, 62.1%), followed by SLE with no other specification (M329) (n = 157, 37.9%), and the average disease evolution was 8.9 ± 6.0 years (range: 1–28 years). A severe SLE manifestation at baseline was found in 174 patients (42.1%), and 80.8% of the patients (n = 334) had at least one recorded comorbidity (Table 1).

The initial SLEDAI report for the 413 patients indicated an average of 1.77 ± 2.5 points on the scale. A majority of cases had no activity (SLEDAI = 0, n = 215, 52.0%), followed by mild activity (SLEDAI 1 to 5, n = 154, 37.3%) and moderate activity (SLEDAI 6 to 10, n = 41, 9.9%); very few cases had high activity (SLEDAI > 10, n = 3, 0.7%). At the next monitoring point for each patient (n = 413), the mean score was 1.72 ± 2.5 points, and the final SLEDAI report of 227 patients with complete records and information presented a mean SLEDAI score of 1.42 ± 2.3 points (for comparison, this subgroup of patients had an initial SLEDAI report of 1.89 ± 2.6 points and 1.69 ± 2.5 points at the second monitoring point). At the end of follow-up, 141 patients (62.1%) had no activity, 69 (30.4%) had mild activity, 15 (6.6%) had moderate activity, and two had high activity (0.9%).

In addition, during the follow-up year, incidental SLE manifestations were found in 45 patients (10.9%), with higher frequencies of lupus nephritis (n = 14, 3.4%), anemia (n = 8, 1.9%), vasculitis (n = 7; 1.7%), thrombocytopenia (n = 7; 1.7%), pleuritis (n = 6; 1.5%), neutropenia (n = 5; 1.2%), venous thrombosis (n = 3, 0.7%), pericarditis (n = 2; 0.5%), and arterial thrombosis, cerebritis, and psychotic episodes, each with one case (n = 1, 0.2%). The mean time to first incident severe SLE manifestation was 346 days. According to disease activity, only 6.0% (n = 13) of those without activity (SLEDAI = 0) had these incident SLE manifestations, compared to those with mild (n = 23, 14.9%) and moderate to high (n = 9, 20.4%) activity.

All patients were being treated with medications on the index date: 129 with monotherapy (31.2%) and the remaining 284 with combination therapy (68.8%). The most frequently used medications, alone or in combination, were chloroquine, hydroxychloroquine, azathioprine, and systemic corticosteroids (Table 2). Seventy-five patients (18.2%) were receiving first-line treatment drugs, 293 (70.1%) were receiving second-line drugs, and 43 (10.4%) were receiving biological third-line drugs, consistent with the finding that 10.9% of the patients had moderate or high activity at the beginning of the cohort.

During the follow-up year, 77 patients (18.6%) started a new medication in addition to the therapy they had been receiving, of which nine started two or more drugs. In addition, 41 patients had their treatment suspended, with two or more medications withdrawn in two patients (Table 3). Fig 1 shows the main sequences of treatments with disease-modifying anti-rheumatic drugs (DMARDs).

Due to the specialized care of IPS-E, 98.1% of patients (n = 405) had a rheumatology visit during the observation year, with a mean of 2.99 ± 1.25 visits per patient per year (range: 1–9, median: 3, interquartile range [IQR]: 2–4). A general practitioner evaluated 185 (44.8%) of the patients, with a mean of 0.73 ± 1.03 (range: 0–6; median: 0 and IQR: 0–1 visits). In addition, 84 diagnostic images or other procedures were performed in 68 different patients, the most frequent being ultrasound (n = 24), mainly abdominal and hepatic, followed by simple X-rays

**Table 1. Clinical and general medical attention characteristics of 413 patients diagnosed with systemic lupus erythematosus treated in a specialized health institution in Colombia.**

| Variable | n = 413 | % | Disease activity—SLEDAI at baseline | | | | | |
|---|---|---|---|---|---|---|---|---|
| | | | No activity (0) | | Mild (1–5) | | Moderate and High (≥ 6) | |
| | | | n = 215 | % | n = 154 | % | n = 44 | % |
| Age (years), mean (SD) | 42.0 (14.2) | | 45.3 (14.4) | | 39.8 (13.8) | | 33.6 (10.0) | |
| Age groups | | | | | | | | |
| 18–44 years | 242 | 58.6 | 110 | 51.2 | 96 | 62.3 | 36 | 81.8 |
| 45–64 years | 148 | 35.8 | 87 | 40.5 | 53 | 34.4 | 8 | 18.2 |
| >65 years | 23 | 5.6 | 18 | 8.4 | 5 | 3.2 | 0 | 0.0 |
| Female | 361 | 87.4 | 186 | 86.5 | 137 | 89.0 | 38 | 86.4 |
| **SLE manifestations at baseline** | | | | | | | | |
| Lupus nephritis | 105 | 25.4 | 48 | 22.3 | 45 | 29.2 | 12 | 27.3 |
| Thrombocytopenia | 25 | 6.1 | 11 | 5.1 | 10 | 6.5 | 4 | 9.1 |
| Pleuritis | 19 | 4.6 | 8 | 3.7 | 7 | 4.5 | 4 | 9.1 |
| Anemia | 14 | 3.4 | 5 | 2.3 | 6 | 3.9 | 3 | 6.8 |
| Neutropenia | 12 | 2.9 | 7 | 3.3 | 4 | 2.6 | 1 | 2.3 |
| Vasculitis | 9 | 2.2 | 5 | 2.3 | 3 | 1.9 | 1 | 2.3 |
| Venous Thrombosis | 9 | 2.2 | 5 | 2.3 | 3 | 1.9 | 1 | 2.3 |
| Pericarditis | 8 | 1.9 | 3 | 1.4 | 4 | 2.6 | 1 | 2.3 |
| Cerebritis | 5 | 1.2 | 3 | 1.4 | 2 | 1.3 | 0 | 0.0 |
| Seizures | 4 | 1.0 | 2 | 0.9 | 1 | 0.6 | 1 | 2.3 |
| Psychosis | 2 | 0.5 | 1 | 0.5 | 1 | 0.6 | 0 | 0.0 |
| Raynaud's syndrome | 2 | 0.5 | 1 | 0.5 | 1 | 0.6 | 0 | 0.0 |
| Other manifestations [a] | 10 | 2.2 | 7 | 3.3 | 3 | 1.9 | 0 | 0.0 |
| **Comorbidities at baseline** | | | | | | | | |
| Antiphospholipid syndrome | 95 | 21.8 | 45 | 20.9 | 37 | 24.0 | 13 | 29.5 |
| Hypertension | 76 | 18.4 | 40 | 18.6 | 31 | 20.1 | 5 | 11.4 |
| Hypothyroidism | 75 | 18.1 | 42 | 19.5 | 28 | 18.2 | 5 | 11.4 |
| Sjögren's syndrome | 54 | 13.0 | 30 | 14.0 | 21 | 13.6 | 3 | 6.8 |
| Osteoporosis | 52 | 12.6 | 34 | 15.8 | 17 | 11.0 | 1 | 2.3 |
| Dyslipidemia | 46 | 11.1 | 23 | 10.7 | 16 | 10.4 | 7 | 15.9 |
| Fibromyalgia | 46 | 11.1 | 25 | 11.6 | 17 | 11.0 | 4 | 9.1 |
| Arthritis | 26 | 6.3 | 19 | 8.8 | 4 | 2.6 | 3 | 6.8 |
| Renal failure | 24 | 5.8 | 8 | 3.7 | 13 | 8.4 | 3 | 6.8 |
| Type 2 diabetes mellitus | 22 | 5.3 | 9 | 4.2 | 9 | 5.8 | 4 | 9.1 |
| Depression | 21 | 5.1 | 5 | 2.3 | 12 | 7.8 | 4 | 9.1 |
| Osteoarthritis | 21 | 5.1 | 15 | 7.0 | 5 | 3.2 | 1 | 2.3 |
| Hearth failure | 4 | 1.0 | 1 | 0.5 | 2 | 1.3 | 1 | 2.3 |
| Chronic Obstructive Pulmonary Disease | 4 | 1.0 | 2 | 0.9 | 2 | 1.3 | 0 | 0.0 |
| **Medical attention** | | | | | | | | |
| General practitioner (number of visits) | | | | | | | | |
| Mean (SD) | 0.73 (1.03) | | 0.62 (0.93) | | 0.79 (1.11) | | 1.09 (1.16) | |
| Count | 185 | 44.8 | 87 | 40.5 | 71 | 46.1 | 27 | 61.4 |
| Rheumatology (number of visits) | | | | | | | | |
| Mean (SD) | 2.99 (1.25) | | 3.00 (1.22) | | 2.94 (1.29) | | 3.11 (1.26) | |
| Count | 405 | 98.1 | 212 | 98.6 | 151 | 98.1 | 42 | 95.5 |
| Diagnostic images/procedures | 68 | 16.5 | 42 | 19.5 | 20 | 13.0 | 6 | 13.6 |
| Ultrasounds | 24 | 5.8 | 14 | 6.5 | 7 | 4.5 | 3 | 6.8 |

*(Continued)*

**Table 1.** (*Continued*)

| Variable | n = 413 | % | Disease activity—SLEDAI at baseline | | | | | |
| --- | --- | --- | --- | --- | --- | --- | --- | --- |
| | | | No activity (0) | | Mild (1–5) | | Moderate and High (≥ 6) | |
| | | | n = 215 | % | n = 154 | % | n = 44 | % |
| Simple X-rays | 14 | 3.4 | 8 | 3.7 | 4 | 2.6 | 2 | 4.5 |
| Echocardiography | 12 | 2.9 | 7 | 3.3 | 5 | 3.2 | 0 | 0.0 |
| CAT scans | 10 | 2.4 | 7 | 3.3 | 3 | 1.9 | 0 | 0.0 |

[a] Other manifestations: cutaneous, splenomegaly, pneumonitis, alopecia, polyneuropathy, telangiectasia

SD: standard deviation; CAT scans: computed axial tomography.

**Table 2. Antirheumatic drugs, doses and intervals of use by a cohort of 413 patients diagnosed with systemic lupus erythematosus, Colombia.**

| Drug | n = 413 | % | Mean dose | Unit | Most used interval |
| --- | --- | --- | --- | --- | --- |
| **Conventional Therapy** | | | | | |
| *Chloroquine* | 217 | 52.5 | 148.2 | mg/day | Daily |
| *Hydroxychloroquine* | 128 | 31.0 | 246.2 | mg/day | Daily |
| *Azathioprine* | 187 | 45.3 | 91.1 | mg/day | Daily |
| *Mycophenolate* | 83 | 20.1 | 1717.2 | mg/day | Daily |
| *Methotrexate* | 89 | 21.5 | 15.9 | mg/week | Weekly |
| *Cyclophosphamide* | 28 | 6.8 | 509.1 | mg/month | Monthly |
| *Cyclosporine* | 33 | 8.0 | 92.2 | mg/day | Daily |
| *Leflunomide* | 20 | 4.8 | 20 | mg/day | Daily |
| **Biologic therapy** | | | | | |
| *Rituximab* | 39 | 9.4 | 950 | mg/dose | By protocol |
| *Belimumab* | 3 | 1.0 | 400 | mg/month | Monthly |
| *Infliximab* | 1 | 0.2 | 300 | mg/dose | By protocol |
| *Immunoglobulin* | 1 | 0.2 | 20000 | mg/dose | By protocol |
| **Corticosteroids** | 293 | 70.9 | | | |
| *Prednisolone* | 237 | 57.4 | 10.1 | mg/day | Daily |
| *Deflazacort* | 47 | 11.4 | 9 | mg/day | Daily |
| *Methylprednisolone (oral)* | 4 | 0.9 | 4 | mg/day | Daily |
| *Methylprednisolone (intravenous)* | 17 | 4.1 | 400 | mg/dose | By pulses |
| *Betamethasone* | 2 | 0.4 | 4 | mg/day | Daily |
| **Most used therapy combinations** | | | | | |
| Antimalarial + one immunosuppressant | 121 | 29.4 | | | |
| Antimalarial monotherapy | 74 | 17.9 | | | |
| Immunosuppressant monotherapy | 29 | 7.0 | | | |
| Antimalarial + DMARDs | 37 | 9.0 | | | |
| DMARD monotherapy | 12 | 2.9 | | | |
| Other 91 different combinations | 140 | 33.8 | | | |

DMARDs: disease-modifying antirheumatic drugs

**Table 3. Modifications of treatment for additions or suspensions of medications in the cohort of 413 patients diagnosed with systemic lupus erythematosus in Colombia.**

| Drug | n = 413 | % | Drug | n = 413 | % |
|---|---|---|---|---|---|
| **DMARD started during follow-up** | | | **DMARD suspended during follow-up** | | |
| Azathioprine | 17 | 4.1 | Azathioprine | 14 | 3.4 |
| Mycophenolate | 17 | 4.1 | Chloroquine | 12 | 2.9 |
| Hydroxychloroquine | 6 | 1.5 | Hydroxychloroquine | 4 | 1.0 |
| | | | Methotrexate | 3 | 0.7 |
| Chloroquine | 5 | 1.2 | Cyclophosphamide | 2 | 0.5 |
| Methotrexate | 5 | 1.2 | Leflunomide | 1 | 0.2 |
| Leflunomide | 4 | 1 | Mycophenolate | 1 | 0.2 |
| Cyclophosphamide | 3 | 0.7 | | | |
| Cyclosporine | 2 | 0.5 | | | |
| **Corticosteroids started during follow-up** | | | **Corticosteroids suspended during follow-up** | | |
| Methylprednisolone | 8 | 1.9 | Prednisolone | 3 | 0.7 |
| Prednisolone | 6 | 1.5 | Deflazacort | 1 | 0.2 |
| Deflazacort | 4 | 1 | | | |
| Betamethasone | 1 | 0.2 | | | |
| **Biologic drugs started during follow-up** | | | **Biologic drugs suspended during follow-up** | | |
| Rituximab | 7 | 1.7 | Rituximab | 1 | 0.2 |
| Belimumab | 1 | 0.2 | Belimumab | 1 | 0.2 |

DMARDs: disease-modifying antirheumatic drugs

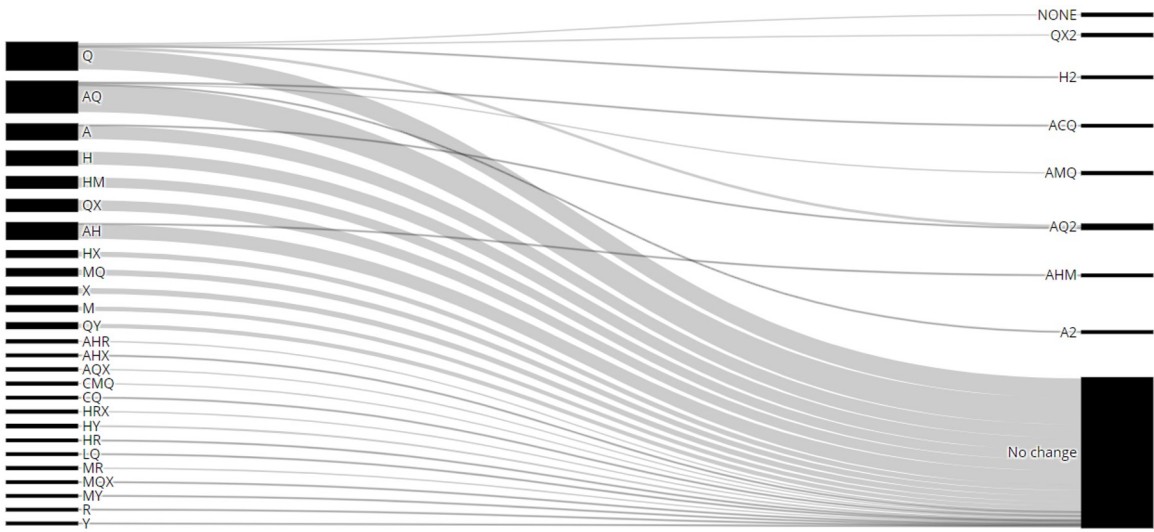

**Fig 1. Disease-modifying antirheumatic drug (DMARD) treatment sequence considering the first and last prescription during follow-up.** Only combinations with at least two patients were included (n = 338). Azathioprine (A), cyclophosphamide (C), cyclosporine (Y), chloroquine (Q), hydroxychloroquine (H), leflunomide (L), methotrexate (X), mycophenolate (M), and rituximab (R). The number 2 on the right refers to final therapies.

**Table 4. Annual direct costs associated with the treatment of 413 patients diagnosed with SLE, Colombia.**

| Direct costs | Proportion of total cost (%) | Mean cost (USD) | Total cost (USD) |
|---|---|---|---|
| **Total Costs—Direct care and treatment cost (annual)** | | | |
| *Total annual cost (n = 413)* | **100.0** | $1,954 | $807,018.6 |
| *DMARDs* | **79.5** | $1,555 | $642,013.3 |
| *Conventional DMARDs annual cost (n = 370)* | **29.8** | $516 | $191,066.2 |
| *Biologic therapy annual cost (n = 43)* | **70.2** | $10,487 | $450,947.1 |
| *Mean cost in patients treated with rituximab* | - | $11,049 | $430,928.1 |
| *Mean cost in patients treated with belimumab* | - | $5,133 | $15,397.5 |
| *Mean cost in patients treated with infliximab* | - | $4,621 | $4,621.5 |
| *Corticosteroids total cost* | **1.8** | $36 | $14,789.4 |
| *Analgesics total cost* | **1.6** | $33 | $13,455.2 |
| *Annual cost of laboratory tests and follow-up* | | | |
| *Cost of medical consultations (n = 413)* | **4.4** | $86 | $35,546.4 |
| *Cost of drugs infusions (n = 43)* | **1.3** | $235 | $10,114.7 |
| *Cost of diagnostic and paraclinical images (n = 413)* | **1.2** | $22 | $9,018.0 |
| *Cost of paraclinical follow-up of SLE (n = 413)* | **10.2** | $199 | $82,081.5 |

* Laboratory tests included antibody measurements, blood counts, and creatinine.

DMARDs: disease-modifying antirheumatic drugs. SLE: systemic lupus erythematosus. USD: United States dollars

(n = 15), mainly of the thorax, echocardiograms (n = 12) and computed axial tomography (CAT) scans (n = 10), among other different paraclinical exams (Table 1).

## Costs associated with care

The main source of costs was pharmacological therapy (82.9% of total costs), especially in those patients who received biological therapy, followed by costs related to paraclinical monitoring (10.2%), such as measuring antibodies and other laboratory tests (see Table 4).

## Discussion

This study described the clinical characteristics of a cohort of patients with SLE undergoing treatment in a rheumatology clinic (IPS-E in Colombia), including pharmacological management over a year of follow-up, the resources used and the costs associated with their outpatient care. The findings establish that SLE in Colombia is a condition that carries a significant disease burden for both patients and health system actors based on the high need for medical care, use of medications, related complications and costs.

The prevalence of SLE in women between 40 and 50 years of age and an average disease evolution of approximately 10 years are similar to findings in other cohorts of patients with SLE [20–24]. These and similar studies show how the frequency of manifestations such as lupus nephritis that are related to significant adverse quality of life changes are a reflection of increased disease activity and severity and even greater mortality [21, 25, 26]. However, the present cohort of patients had lower disease severity and fewer complications than a cohort in Japan that had a prevalence of lupus nephritis of 41.2% (90.5% with moderate or high activity). In that study, however, severity was estimated based on the medication used, while in this study, it was obtained through the validated SLEDAI score [20]. Similarly, in the study by Kent et al., 44.6% of the patients had moderate or severe activity, but the measurement was conducted online through a self-reported activity [21].

Approximately 11% of patients had some new manifestations during follow-up, which may be evidence that uncontrolled patients with moderate or high activity may develop complications that require comprehensive care, such as adequate and close specialized monitoring by rheumatology and other medical specialties and access to diagnostic paraclinical tests and appropriate pharmacological therapies, including immunomodulatory or biotechnological drugs when indicated [27, 28].

Another critical point is the coexistence of comorbidities, mainly antiphospholipid syndrome, Sjogren's syndrome, and some cardiovascular and musculoskeletal disorders, similar to those reported in the United Kingdom and the United States [29, 30]. Comorbidities can directly impact survival; for example, Ocampo-Piraquive et al. [26] stated that patients with SLE have twice the risk of cardiovascular mortality as healthy controls of the same age and sex. This relationship highlights the need for better control of associated risk factors to reduce premature deaths, which can be achieved through multidisciplinary efforts [31]. This situation is present in both developed and developing countries; however, data on Latin American countries are scarce [26]. Depression (5.1%) and fibromyalgia (11.1%) were also notable in the evaluated cohort. These conditions occur frequently in multiple groups of patients with SLE and have been described as important predictors of reduced quality of life and greater work disability and disease activity [7, 25, 32].

The vast majority of patients were receiving standard of care treatment (antimalarials and corticosteroids), which has been recommended for decades as an effective therapy in those with cutaneous and hematologic manifestations of SLE [1]. The greater use of chloroquine and hydroxychloroquine is noteworthy. The latter is preferable due to its lower risk of retinal toxicity; however, given that the cost is several times higher, its use may be limited in Colombia. Second-line therapy with immunomodulators was used in more than two thirds of the patients, which corresponds to the disease activity and frequency of complication. However, this type of medication is associated with a higher frequency of adverse reactions and adherence issues [33, 34]. The higher prescription of rituximab compared with belimumab may reflect the approval of the latter in Colombia at the end of 2014. Consequently, belimumab might not have been broadly diffused and available at the time of the study.

In addition, more than 20% of the patients received methotrexate, a drug not approved in Colombia for this indication, although clinical practice guidelines mention its potential usefulness and other studies have shown its efficacy in reducing SLEDAI and the dose of corticosteroids [1, 35]. Approximately 70% of the patients had at least one corticosteroid prescription; there are no previous data on the prevalence of corticosteroid use in Colombian patients with SLE, but this prevalence is slightly higher than those reported in other SLE cohorts, such as the prevalence of 60% reported by Kan H et al. in a US study [36].

The costs associated with the care of patients with SLE have been poorly studied in Colombia and Latin America in general, but in studies conducted in North America and Europe, an annual direct cost range of USD 3,735–14,410 has been established. A study of Medicaid patients estimated an incremental cost of USD 10,984 in 2012 [30], while another study in the United Kingdom estimated the cost at approximately USD 2,898 [29]. Establishing the cost of the services provided to each patient, including medications, is essential. In the present study, the estimated direct annual cost was USD 1,954, which is less than in developed countries but quite high for the average cost paid by Colombian health insurers. The amount allocated to the care of a general Colombian patient would be on average USD 255 annually, which is approximately 8 times less than the cost of a patient with SLE [37]. In a study carried out in Colombia by Prada et al. with data directly provided by 2 insurers, they found an annual average direct cost of USD 2,355 between 2014 and 2015, similar to the current study, although they included data on hospital care and a lower exchange rate [14]. Changes in quality of life and loss of

productivity of the patient and their family, which were not estimated in this study, can represent a very significant economic burden for society and can vary between USD 1,093 and 14,610 in North America and Europe depending on the country and evaluation method [38].

Notably, the direct costs of patients with severe SLE receiving biological therapy exceed USD 10,487 annually, with a range of USD 4,621 to 11,049 depending on the type of medication used (biological therapy). This high therapy cost represents a significant burden on the national health system but should be compared with the costs of severe manifestations and the long-term consequences of disease activity. Patients with higher SLEDAI scores, mainly driven by lupus nephritis (as described in different studies, including a previous study carried out in Colombia [14]), have higher direct costs, which can be a critical point for the primary and secondary care and prevention of SLE [29, 39]. A study in five European countries suggested that the average annual direct costs were USD 2,972 for a patient with low SLEDAI scores but USD 5,326 for those with higher scores [40]. In addition, patients with higher scores visit the emergency department more frequently, which further increases direct care-related costs [32, 41].

Some limitations were identified in this study. Only patients from the Colombian contributory regime were followed; therefore, the conclusions can only be extrapolated to similar health insurance regimes and health care. The information was obtained retrospectively from clinical records; therefore, certain variables were not complete (such as final SLEDAI or the exact date of SLE diagnosis) or available (for example other laboratory test results). It was not possible to establish the frequency of emergency department visits and the costs associated with hospital care because these data were not available in the dispensing database or the ambulatory health records used. Some cost calculations were performed according to fee schedules, which could differ from the real value billed and paid for the services provided. Finally, the follow-up period was relatively short. This is the first pharmacoepidemiological study of SLE clinical characteristics and drug therapy patterns to use clinical records as a data source. Thus, it contributes to the body of knowledge regarding costs of care in Colombia, which is critical for generating evidence and providing a baseline for subsequent studies.

## Conclusions

Based on the findings, it can be concluded that SLE generates a significant morbidity and economic burden on the Colombian health system associated with the costs of medical care, diagnostic and follow-up support, complications, and, in particular, the use of medications when biotech therapy is required. This study presents an approach to determine the direct costs associated with SLE care in the country, but it also generates new questions. We recommend conducting new studies that address other relevant scenarios, such as the rate of exacerbations of these patients, long-term follow-up, direct costs related to hospital care (i.e., emergency department visits, laboratory tests), and indirect costs associated with the disease. Such studies may ultimately help improve the care of SLE patients in this country and in other healthcare systems. Furthermore, these and future results may enable better decisions on the rational use of pharmaceutical and technological resources to achieve the best clinical outcomes.

## Acknowledgments

The authors gratefully acknowledge Soffy López for his support in obtaining the initial database of patients.

## Author Contributions

**Conceptualization:** Jorge Enrique Machado-Alba, Manuel E. Machado-Duque, Carolina Duarte-Rey, Andrés González-Rangel.

**Funding acquisition:** Jorge Enrique Machado-Alba, Andrés González-Rangel.

**Investigation:** Jorge Enrique Machado-Alba, Manuel E. Machado-Duque, Andres Gaviria-Mendoza.

**Methodology:** Jorge Enrique Machado-Alba, Manuel E. Machado-Duque, Andres Gaviria-Mendoza.

**Project administration:** Jorge Enrique Machado-Alba.

**Resources:** Jorge Enrique Machado-Alba.

**Supervision:** Jorge Enrique Machado-Alba.

**Validation:** Jorge Enrique Machado-Alba, Manuel E. Machado-Duque, Andres Gaviria-Mendoza.

**Writing – original draft:** Manuel E. Machado-Duque, Andres Gaviria-Mendoza.

**Writing – review & editing:** Jorge Enrique Machado-Alba, Manuel E. Machado-Duque, Andres Gaviria-Mendoza.

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
