## [Decision Letter · Decision Letter 0]

13 Jul 2022

PONE-D-22-14695Clinical characterization of a cohort of patients under treatment for systemic lupus erythematosus in ColombiaPLOS ONE

Dear Dr. Machado-Alba,

Thank you for submitting your manuscript to PLOS ONE. After careful consideration, we feel that it has merit but does not fully meet PLOS ONE’s publication criteria as it currently stands. Therefore, we invite you to submit a revised version of the manuscript that addresses the points raised during the review process.

We look forward to receiving your revised manuscript.

Kind regards,

Nasser Hadal Alotaibi

Academic Editor

PLOS ONE

Journal Requirements:

This study received funding from GSK. The funder had the following involvement with the study: GSK did not participate in the management or data analysis of the study information.   CDR and AGR are GSK employees; JMA, AGG and MMD have developed studies funded by GSK, Pfizer, Biotoscana, Sanofi, Sanofi-Pasteur, Abbot, Tecnoquímicas, Bayer, Novartis. 

Reviewers' comments:

Reviewer's Responses to Questions

**Comments to the Author**

1. Is the manuscript technically sound, and do the data support the conclusions?

Reviewer #1: Partly

Reviewer #2: Yes

2. Has the statistical analysis been performed appropriately and rigorously? 

Reviewer #1: I Don't Know

Reviewer #2: Yes

3. Have the authors made all data underlying the findings in their manuscript fully available?

Reviewer #1: Yes

Reviewer #2: Yes

4. Is the manuscript presented in an intelligible fashion and written in standard English?

Reviewer #1: Yes

Reviewer #2: No

5. Review Comments to the Author

Reviewer #1: General comments

There are a lot of abbreviations where they were not full explained before they first appear

The methods part is not well written; I believe this needs to be majorly revised again

The paper overall needs major revision with a grammars and needs to be clearly written

Introduction

It is well written and the background and statement of the problem is given well

Methods

ICD- mention this first before indicating the abbreviation applies to others too

Design and patients section is not well written please revise this paragraph.

Why did you include all patients irrespective of age? (any implication you see on the result)? Does SLE not differ by age? Very young and very old? How do you see their comparison?

Why did you remove those who were only 3 month diagnosed/or cared? Give justification

Why did you only include 12 months? This seems short period? Any implication on your result

What do you mean by the data was monitored for one year? (re-write this sentence).

Why have not applied ICD 11 instead of 10?

On line 119 what do you mean by trained physician? Specify it very well? Make a detail explanation how your data collectors were trained for your study? Was there a supervisor?

On line 150 can you mention a reference?

The sample size calculation is not clear at all? How can you insure representativeness? Where did you get the estimate? It needs a reference

Line 161-162 is not clear. Please clearly state the methods used to test normality

If you use Kaplan mayor this study cannot be a descriptive study…

The methods lack quality assurance, tools used for data collection, what did you do with missing data? How did you clean the data?

Results

Any statement you want to mention about incomplete data you found? How did you handle it? How many did you review (how many data was available in the mentioned duration)

What you wrote on the first paragraph of the result and on the sample size determination is different. Did you have 373 or 413 sample size?

On costs associated with care; can you mention the frequency and percentage? Line number 236’

You have not reported anything related with Kaplan-Meier survival result? If you did please specify very well…

Line 326 what do you mean by certain cases? Can you quantify?

Line 327 why was it not possible to establish the frequency of emergency department visits, as well as the costs associated with hospital care.

Line 328 why were costs calculated according to fee schedules?

Line 329 why didn’t you make the follow up period longer then?

Conclusion

Line 337 Why did you mention a reference here. the concussion part should be your own words no reference is needed here

What is DMARD?

Reviewer #2: In this manuscript, the authors aimed to assess the clinical characteristics and health care resource utilization in a Colombian SLE outpatient cohort. The study is interesting, but a number of issues should be addressed before its publication.

Major issues

1. How was sample size calculated? Try to elaborate the method used for calculation of the sample size.

2. Sampling design used in the study was not specified. Try to specify how the patients was selected from the source population.

3. I did not see any recommendation in the main text. Thus, a ‘recommendation’ should be added after the ‘conclusion’ section of the main text.

Minor issues

1. The full form of the abbreviated word ‘SLEDAI’ should be properly written.

2. I suggest the authors’ to add a recommending sentence within the conclusion part of the abstract.

3. I suggest the authors’ to rename the sub-section ‘Design and patients’ within the method section of the main text to ‘Study design and setting’. Moreover, a new sub-section ‘source and study population’ should be added (this can include the study population and the exclusion criteria which is already explained within the sub-section ‘Design and patients’).

4. Try to add the main statistical tools used in the study in the method of the abstract section.

5. Ethical approval sound appropriate. Please detail ethical approval body with address. Add Reference number of approval letter.

6. Most of your findings were compared with the findings of developed countries only. I also recommend the authors’ to compare their findings with that of the developing countries.

7. I suggested the title of the article to be renamed as “Assessment of clinical characterization of a cohort of patients under treatment for systemic lupus erythematosus in Colombia: a retrospective study”.

8. Referencing style need adjustment. It should be in compliance with the PLOSONE guidelines.

9. Your manuscript does not follow PLOS ONE’s manuscript preparation guidelines. Try to revisit the guidelines and adjust the whole manuscript accordingly.

10. Coherence and readability of this manuscript need improvement. English editing is required.

6. PLOS authors have the option to publish the peer review history of their article (what does this mean?). If published, this will include your full peer review and any attached files.

Reviewer #1: **Yes: **Tsion Afework

Reviewer #2: No

---

## [Author Response · Author response to Decision Letter 0]

10 Aug 2022

PONE-D-22-14695R1

Clinical characterization of a cohort of patients under treatment for systemic lupus erythematosus in Colombia: a retrospective study

Dear Edrian Nim Tolentino

PLOS ONE

We have reviewed all your concerns and we respond to each of them.

1. Thank you for stating the following in the Competing Interests section:

This study received funding from GSK. The funder had the following involvement with the study: GSK did not participate in the management or data analysis of the study information. CDR and AGR are GSK employees; JMA, AGG and MMD have developed studies funded by GSK, Pfizer, Biotoscana, Sanofi, Sanofi-Pasteur, Abbot, Tecnoquímicas, Bayer, Novartis.

Response/ Thank you.

Response/ confirmed, we add “This does not alter our adherence to PLOS ONE policies on sharing data and materials”

Response/ confirmed, we add “This does not alter our adherence to PLOS ONE policies on sharing data and materials” in cover letter. 

2. Please provide additional details regarding participant consent. In the Methods section, please ensure that you have specified (1) whether consent was informed and (2) what type you obtained (for instance, written or verbal). If your study included minors, state whether you obtained consent from parents or guardians. If the need for consent was waived by the ethics committee, please include this information.

Response/ we add in Ethics statement

According to Colombian regulations (Decree 8430 of 1993), risk-free research that takes information from clinical records does not require informed consent. The research was endorsed by the Bioethics Committee of the Universidad Tecnológica de Pereira in the risk-free category (Approval code: CBE 132018). The principles established by the Declaration of Helsinki were respected. No personal data from any of the research participants were considered.

3. We note your current Data Availability statement is:

"Yes - all data are fully available without restriction"

"This text is appropriate if the data are owned by a third party and authors do not have permission to share the data."

Response/ ok, we will make the correction in the journal application.

We also note you included the following, additional information in your response to reviewers:

"The original contributions presented in the study are included in the article/supplementary material, further inquiries can be directed to the corresponding author/s." The name of the information repository is: protocols.io

Code availability: dx.doi.org/10.17504/protocols.io.bzufp6tn”

PLOS journals require authors to make the minimal data set publicly available without restriction at the time of publication. PLOS defines the “minimal data set” as consisting of the data set used to reach the conclusions drawn in the manuscript with related metadata and methods, and any additional data required to replicate the reported study findings in their entirety. This includes:

Can you please address the following and clarify where the relevant data are located:

a.) Can you please clarify whether the minimal dataset for this study are uploaded to protocols.io and available via the provided DOI (dx.doi.org/10.17504/protocols.io.bzufp6tn)?

Response/ Yes, the minimal dataset for this study are uploaded to protocols.io and available via the provided DOI (dx.doi.org/10.17504/protocols.io.bzufp6tn)

b.) If not, are the data third party data (i.e., data not owned or collected by the author(s))? If these are indeed third party data, please explain how others can access these datasets and confirm that others would be able to access these data in the same manner as the authors. Please also confirm that the authors did not have any special access privileges that others would not have.

Response/ All data is owned by the authors. In addition, the data was correctly anonymized. as it says in methods, where it is indicated " The principles established by the Declaration of Helsinki were respected. No personal data from any of the research participants were considered”.

c.) If not, but there are ethical or legal restrictions on sharing a de-identified data set, please explain them in detail (e.g., data contain potentially identifying or sensitive information) and who has imposed them (e.g., a governmental body, an ethics committee, etc.). Please also provide non-author contact information* for a data access committee, ethics committee, or other institutional body to which data requests may be sent.

Response/ N/A

* PLOS’ Data Policy states that authors cannot be the sole named individuals responsible for ensuring data access (https://journals.plos.org/plosone/s/data-availability#loc-unacceptable-data-access-restrictions).

Response/ The anonymized data are public since the manuscript was written and are available at the corresponding link for anyone who wants to consult them.

We will update your Data Availability statement on your behalf using the information you provide.

The authors

---

## [Decision Letter · Decision Letter 1]

6 Oct 2022

PONE-D-22-14695R1Clinical characterization of a cohort of patients under treatment for systemic lupus erythematosus in Colombia: a retrospective studyPLOS ONE

Dear Dr. Machado-Alba,

Thank you for submitting your manuscript to PLOS ONE. After careful consideration, we feel that it has merit but does not fully meet PLOS ONE’s publication criteria as it currently stands. Therefore, we invite you to submit a revised version of the manuscript that addresses the points raised during the review process.

We look forward to receiving your revised manuscript.

Kind regards,

Nasser Hadal Alotaibi

Academic Editor

PLOS ONE

Journal Requirements:

Reviewers' comments:

Reviewer's Responses to Questions

**Comments to the Author**

1. If the authors have adequately addressed your comments raised in a previous round of review and you feel that this manuscript is now acceptable for publication, you may indicate that here to bypass the “Comments to the Author” section, enter your conflict of interest statement in the “Confidential to Editor” section, and submit your "Accept" recommendation.

Reviewer #1: All comments have been addressed

Reviewer #2: All comments have been addressed

2. Is the manuscript technically sound, and do the data support the conclusions?

Reviewer #1: Yes

Reviewer #2: Yes

3. Has the statistical analysis been performed appropriately and rigorously? 

Reviewer #1: Yes

Reviewer #2: Yes

4. Have the authors made all data underlying the findings in their manuscript fully available?

Reviewer #1: Yes

Reviewer #2: Yes

5. Is the manuscript presented in an intelligible fashion and written in standard English?

Reviewer #1: Yes

Reviewer #2: No

6. Review Comments to the Author

Reviewer #1: Overall the paper is well revised; I just have few comments

On Line 39 and line 101 please re-write again, don’t put full stops before finishing a sentences, re-write it again.

Why didn’t you give any recommendation for the missing variables (or incomplete) like emergency department visits, costs associated with hospital care, ambulatory health records

Do you think you have measured economic burden well? Can you say something about that on the paper?

On the data source can you describe the time frame of your data collection.

Reviewer #2: Minor Comment

-English editing is required. Coherence and readability of the manuscript need an improvement.

7. PLOS authors have the option to publish the peer review history of their article (what does this mean?). If published, this will include your full peer review and any attached files.

Reviewer #1: No

Reviewer #2: **Yes: **Nuru Abdu

---

## [Author Response · Author response to Decision Letter 1]

18 Oct 2022

Response to reviewers

Dear Editor

Journal: PLOS ONE

Manuscript ID number: PONE-D-22-14695R1

Title: Clinical characterization of a cohort of patients treated for systemic lupus erythematosus in Colombia: a retrospective study

We have reviewed the manuscript and responded to each concern. In addition, AJE, who performed the translation, was asked to review the coherence and readability.

Reviewer #1: Overall the paper is well revised; I just have few comments

On Line 39 and line 101 please re-write again, don’t put full stops before finishing a sentences, re-write it again.

Response: We corrected the sentences on lines 39 and 101 regarding the study design.

Why didn’t you give any recommendation for the missing variables (or incomplete) like emergency department visits, costs associated with hospital care, ambulatory health records

Response: In the conclusions section, we added a sentence recommending future studies that include these variables. Ambulatory health records were the main source of information in the study.

Do you think you have measured economic burden well? Can you say something about that on the paper?

Response: We think that economic burden was adequately measured using the proposed methodology. The limitations regarding cost calculations were addressed. We think it is unnecessary to further discuss this issue in the paper.

On the data source can you describe the time frame of your data collection.

Response: The data frame is described in the subsection “Study design and settings” and corresponded to July 1, 2016, through June 30, 2017.

Reviewer #2: Minor Comment

-English editing is required. Coherence and readability of the manuscript need an improvement.

Response: The final version of the manuscript was edited by American Journal Experts. We hope this editing addresses this comment. We add the certificate.

The authors

---

## [Decision Letter · Decision Letter 2]

9 Feb 2023

PONE-D-22-14695R2Clinical characterization of a cohort of patients treated for systemic lupus erythematosus in Colombia: a retrospective studyPLOS ONE

Dear Dr. Machado-Alba,

Thank you for submitting your manuscript to PLOS ONE. After careful consideration, we feel that it has merit but does not fully meet PLOS ONE’s publication criteria as it currently stands. Therefore, we invite you to submit a revised version of the manuscript that addresses the points raised during the review process.

We look forward to receiving your revised manuscript.

Kind regards,

Nasser Hadal Alotaibi

Academic Editor

PLOS ONE

Journal Requirements:

Reviewers' comments:

Reviewer's Responses to Questions

**Comments to the Author**

1. If the authors have adequately addressed your comments raised in a previous round of review and you feel that this manuscript is now acceptable for publication, you may indicate that here to bypass the “Comments to the Author” section, enter your conflict of interest statement in the “Confidential to Editor” section, and submit your "Accept" recommendation.

Reviewer #1: (No Response)

Reviewer #2: All comments have been addressed

2. Is the manuscript technically sound, and do the data support the conclusions?

Reviewer #1: Yes

Reviewer #2: Yes

3. Has the statistical analysis been performed appropriately and rigorously? 

Reviewer #1: No

Reviewer #2: Yes

4. Have the authors made all data underlying the findings in their manuscript fully available?

Reviewer #1: Yes

Reviewer #2: Yes

5. Is the manuscript presented in an intelligible fashion and written in standard English?

Reviewer #1: Yes

Reviewer #2: Yes

6. Review Comments to the Author

Reviewer #1: Review for : Clinical characterization of a cohort of patients treated for systemic lupus erythematosus in Colombia: a retrospective study

The authors have addressed all comments given last time and has revised the manuscript.

Just minor comments.

Abstract

I wouldn’t advise use of abbreviation on the abstract section of manuscript

Methods

Does the outcome of SLE not different according to age? How did you take account of that during your analysis? Since you included all age? But on the result it says 18-86..so you should either say we included all ages that are 18 and above

Your sample size is totally not clear? It needs reference, show the formula you used, what does expect frequency 50% mean??? For retrospective cohort study there is a formula ..

On the ethical declaration section good to include reference for Colombian regulations (Decree 8430 of 1993)

Result

Either report the age result in mean and SD or with the category mentioning both at the same time is not necessary.

What I understood is this is observation retrospective study or retrospective cohort study do please report the incidence of SLE don’t talk about prevalence

Limitation

How does excluding incomplete data such as clinometric cases affected or limited your study in any way?

How many cards did you initially find? And how many of them were incomplete? Mention this in the result first section I think this information is relevant to mention as limitation too…you said some variables are incomplete so what is that implication on your finding.. what were the variables

Reviewer #2: I would like to appreciate the authors for taking their time to correctly address the issues raised during peer-review process. All the comments are satisfactorily addressed. This paper is a candidate for publication.

7. PLOS authors have the option to publish the peer review history of their article (what does this mean?). If published, this will include your full peer review and any attached files.

Reviewer #1: No

Reviewer #2: **Yes: **Nuru Abdu

---

## [Author Response · Author response to Decision Letter 2]

24 Mar 2023

Response to reviewers

Dear Editor

Journal: PLOS ONE

Manuscript ID number: PONE-D-22-14695R2

Title: Clinical characterization of a cohort of patients treated for systemic lupus erythematosus in Colombia: a retrospective study

We have reviewed the manuscript and responded to each concern. 

Comments to the Author

Reviewer #1: 

The authors have addressed all comments given last time and has revised the manuscript.

Just minor comments.

Abstract

I wouldn’t advise use of abbreviation on the abstract section of manuscript

R/ We have avoided the use of systemic lupus erythematosus (SLE) abbreviation. we can remove the acronym and leave the full words.

Methods

Does the outcome of SLE not different according to age? How did you take account of that during your analysis? 

R/ We have now included the results of the SLEDAI in table 1 according to the age categories in order to show the differences according to age. 

Since you included all age? But on the result it says 18-86..so you should either say we included all ages that are 18 and above

R/ Corrected, thank you

Your sample size is totally not clear? It needs reference, show the formula you used, what does expect frequency 50% mean??? For retrospective cohort study there is a formula ..

R/ This is an observational, retrospective study. We used data from a cohort of patients with SLE, but the study design was not an analytical cohort study (we did not have an exposed and unexposed group comparison). An expected frequency of 50% was used to reach the highest sample size. We have now included the formula and references (17 and 18). We changed the term “expect frequency” for “estimated proportion”. 

On the ethical declaration section good to include reference for Colombian regulations (Decree 8430 of 1993)

R/ Included (reference 19)

Result

Either report the age result in mean and SD or with the category mentioning both at the same time is not necessary.

R/ We have eliminated the age groups from the text. 

What I understood is this is an observation retrospective study or retrospective cohort study do please report the incidence of SLE don’t talk about the prevalence

R/ We did not consider the new cases of SLE during the time. We had a database of patients already diagnosed with SLE. Thus, the incidence of SLE can not be calculated. However, we did show the incidence of certain SLE manifestations during the follow-up year, such as lupus nephritis.

Limitation

How does excluding incomplete data such as clinometric cases affected or limited your study in any way?

R/ Only patients with an initial SLEDAI were included in the sample, and thus the results showed in the manuscript for the 413 patients include this clinimetric information. The limitation is for the final SLEDAI results, which were available in only 227 patients. This is already explicitly stated in the second paragraph of the results section and in the limitations (in the discussion section). 

How many cards did you initially find? And how many of them were incomplete? Mention this in the result’s first section I think this information is relevant to mention as a limitation too…you said some variables are incomplete so what is that implication on your finding.. what were the variables

R/ We do not have the data of initial registries that were incomplete. For the variables included in the study, only the last SLEDAI and the exact date of diagnosis had missing values. This is now included in the limitations. When we say that some variables were not complete due to the origin of the data, we also intend to mention that some other variables were not available (data was retrospective). We have now added this in the limitations. 

Reviewer #2

I would like to appreciate the authors for taking their time to correctly address the issues raised during peer-review process. All the comments are satisfactorily addressed. This paper is a candidate for publication.

R/ thank you

The authors

---

## [Editor Report · Decision Letter 3]

4 May 2023

Clinical characterization of a cohort of patients treated for systemic lupus erythematosus in Colombia: a retrospective study

PONE-D-22-14695R3

Dear Dr. Machado-Alba,

We’re pleased to inform you that your manuscript has been judged scientifically suitable for publication and will be formally accepted for publication once it meets all outstanding technical requirements.

Kind regards,

Nasser Hadal Alotaibi

Academic Editor

PLOS ONE
---

## [Editor Report · Acceptance letter]

10 May 2023

PONE-D-22-14695R3 

Clinical characterization of a cohort of patients treated for systemic lupus erythematosus in Colombia: a retrospective study 

Dear Dr. Machado-Alba:

I'm pleased to inform you that your manuscript has been deemed suitable for publication in PLOS ONE. Congratulations! Your manuscript is now with our production department. 

Kind regards, 

on behalf of

Dr. Nasser Hadal Alotaibi 

Academic Editor

PLOS ONE